# Genetic and Functional Effects of Adiponectin in Type 2 Diabetes Mellitus Development

**DOI:** 10.3390/ijms232113544

**Published:** 2022-11-04

**Authors:** Yu-Hui Tang, Yeh-Han Wang, Chin-Chang Chen, Chia-Jung Chan, Fuu-Jen Tsai, Shih-Yin Chen

**Affiliations:** 1Department of Medical Genetics, China Medical University Hospital, Taichung 40447, Taiwan; 2Department of Anatomical Pathology, Taipei Institute of Pathology, School of Medicine, National Yang-Ming University, Taipei 112304, Taiwan; 3Department of Anatomy, School of Medicine, China Medical University, Taichung 40447, Taiwan; 4Genetics Center, Department of Medical Research, China Medical University Hospital, China Medical University, No. 2, Yuh-Der Road, Taichung 40447, Taiwan; 5School of Chinese Medicine, China Medical University, Taichung 40447, Taiwan

**Keywords:** C57BLKsJ-db/db mice, T2DM, adiponectin

## Abstract

Diabetes mellitus (DM) is a common chronic metabolic disease, and the C57BLKsJ-db/db mice are good animal models for type 2 diabetes mellitus (T2DM). In this study, Western blotting and immunohistochemistry (IHC) were employed to examine the protein expression of adiponectin in the liver tissues of T2DM mice with different disease courses (4, 16, and 32 weeks). Adiponectin expression reduced in the liver tissues of T2DM mice in different disease courses. The genotypic and allelic frequencies of the adiponectin gene rs1063538 and rs2241766 single nucleotide polymorphisms (SNPs) in a Taiwanese population (570 T2DM patients and 1700 controls) were investigated. Based on the genetic distribution of the rs2241766 locus, the distribution frequency of the T allele in the T2DM group (72.8%) was higher than in the control group (68.8%). Individuals carrying the G allele had a 0.82-fold greater risk of developing T2DM than individuals carrying the T allele. Differences were evident in the genotypic and allelic distributions (*p* < 0.05). Enzyme-linked immunosorbent assay (ELISA) was used to measure changes in serum adiponectin protein concentrations in the healthy population and in patients with T2DM. Serum adiponectin concentration in patients with T2DM was lower than in the control group. In summary, adiponectin was determined to be a T2DM susceptibility gene and may be involved in T2DM progression.

## 1. Introduction

According to global statistics, the number of adults with diabetes mellitus (DM) was 463 million in 2019 and this figure is expected to reach 700 million by 2045 [1]. Patients with type 2 diabetes mellitus (T2DM) account for approximately 90% of all patients with DM [2,3]. T2DM is a common endocrine and metabolic disorder that is characterized by a poor response to insulin and impaired insulin secretion by pancreatic β-cells [4]. The body’s inability to effectively utilize insulin can hinder the successful entry of blood glucose into tissues and cells, resulting in the abnormal elevation of blood glucose [5]. In general, obesity, poor diet, and lack of exercise are the main causes of T2DM [6,7]. It is worth noting that genetic factors also play an important role in DM [8] and are considered a new pathway by which to address T2DM. Studies have shown that monozygotic twins have a higher probability of developing T2DM than dizygotic twins [9]. The prevalence of DM varies substantially among different ethnicities, even under similar lifestyle environments [10]. In addition, having a family history of T2DM is also an important factor, as offspring with one parent suffering from T2DM have a 40% risk of developing T2DM, and this risk increases to 70% when both parents suffer from T2DM [11].

Adiponectin (ADP) is an endogenous bioactive molecule secreted by adipocytes and it has a molecular weight of 30k Da [12]. The adiponectin gene consists of three exons and two introns, and it encodes for 244 amino acids. This gene is highly expressed in adipose tissue [13]. Adiponectin plays a crucial role in the regulation of carbohydrate and lipid metabolism, and it can promote fatty acid oxidation and carbohydrate utilization to decrease blood glucose and blood lipids [14,15].

The normal range of adiponectin concentration in the blood of healthy individuals is 5 to 30 μg/mL [16], whereas patients with DM, obesity, and coronary heart disease have been found to exhibit substantially lower levels of serum adiponectin than healthy individuals, suggesting that adiponectin level is closely associated with these diseases [17]. Previous research also reported that low-abundant adiponectin receptors in visceral adipose tissue of humans and rats are further reduced in diabetic animals [18]. However, the detailed mechanisms still need further verification. Given the relative lack of studies on adiponectin gene and protein expression in T2DM, a T2DM mouse model (db/db mice) was established to investigate adiponectin expression in order to examine the correlation between adiponectin and T2DM, as well as the effects of adiponectin on T2DM. Furthermore, adiponectin gene and protein expression in murine liver tissues were also investigated to elucidate the role of adiponectin gene and protein expression in T2DM pathology.

The adiponectin gene is located on the chromosome 3q27, and whole-genome scanning of this region identified susceptibility genes for T2DM and metabolic syndrome. Therefore, adiponectin may be involved in the onset of T2DM [19]. The objective of the current study was to determine the single-nucleotide polymorphism (SNP) distribution of locus 45 in exon 2 (rs2241766) [20] and locus 3336 in the 3′-untranslated region (rs1063538) [21] of the adiponectin gene within a T2DM population in Taiwan, and to examine whether SNPs in these two loci can provide meaningful information for T2DM treatment.

ELISA was used to measure changes in serum adiponectin protein between the healthy population and patients with T2DM. Serum adiponectin concentration in T2DM patients was found to be higher than in the control group, and serum adiponectin increased with increasing disease course in patients with T2DM.

Based on the experimental results, the adiponectin gene is believed to be a T2DM susceptibility gene and it may participate in T2DM onset by regulating blood glucose and lipid metabolism. In the future, we hope to examine the pathogenic mechanism of adiponectin in T2DM further to provide a new direction for T2DM treatment.

## 2. Results

### 2.1. Adiponectin Protein Expression in Hepatocytes from Lepr^db^ Mice and Dock^7m^ Mice

Western blotting was performed to compare adiponectin protein expression in hepatocytes from Lepr^db^ mice (T2DM group) and Dock^7m^ mice (control group) with different disease courses (4, 16, and 32 w). There were differences in the adiponectin protein expression in hepatocytes between Lepr^db^ mice (T2DM group) and Dock^7m^ mice (control group) when compared among different disease courses (4, 16, and 32 w) (Figure 1) (liver 4 w, *p* = 0.022; liver 16 w, *p* = 0.028; liver 32 w, *p* = 0.023). Similar results were also observed in the expression level of mRNA (Appendix A).

### 2.2. Adiponectin Expression in Liver Tissues from Lepr^db^ Mice and Dock^7m^ Mice

In order to observe the level of adiponectin protein in murine hepatocytes, immunohistochemistry (IHC) analysis was performed on liver tissue sections from Lepr^db^ mice (T2DM group) and Dock^7m^ mice (control group) to quantify and compare the staining results of mice with different disease courses (4, 16, and 32 w). Adiponectin protein expression was substantially lower in Lepr^db^ mice (T2DM group) at 4, 16, and 32 w compared with Dock^7m^ mice (control group) (Figure 2).

### 2.3. Adiponectin and T2DM Genetic Polymorphism Analysis

The present study obtained the rs1063538 and rs2241766 single-nucleotide polymorphisms (SNPs) at chromosome region 3q27.3 in adiponectin (Figure 3). Table 1 shows the genotypic and allelic distribution of adiponectin gene rs1063538 and rs2241766. The genotypic frequencies of rs1063538 in the control group were: TT 33.8%, TC 49.4%, and CC 16.8%, while those in the T2DM group were: TT 31.8%, TC 50.4%, and CC 17.9%. The percentage of the TT genotype in the control group was higher than in the T2DM group, which implies that this genotype seems to confer a protective effect; the percentage of the CC genotype in the control group was lower than that in the T2DM group, and hence, this genotype appears to be a risk factor. In addition, the frequency of the C allele in rs1063538 was higher in the T2DM group than in the control group (43.1% vs. 41.5%). The percentage of participants carrying the C allele who suffered from T2DM was 1.06-fold higher than that of patients carrying the T allele, although the data do not present statistically significant difference.

The genotypic frequencies of rs2241766 in the control group were: TT 47.5%, TC 42.3%, and CC 10.2%, while those in the T2DM group were: TT 52.5%, TG 40.6%, and GG 6.9%. The percentage of the TT genotype in the control group was lower than that in the T2DM group, and hence, this genotype appears to be a risk factor. The percentage of the GG genotype in the control group was higher than that in the T2DM group, which implies that this genotype seems to confer a protective effect. The frequency of the T allele in rs2241766 was higher in the T2DM group than the control group (72.8% vs. 68.8%). The percentage of patients carrying the G allele who suffered from T2DM was 0.82-fold higher than that of patients carrying the T allele. Genotypic (*p* = 0.025) and allelic frequencies (*p* = 0.009) of rs2241766 presented statistically significant difference.

### 2.4. Serum Adiponectin Expression in the Human Control and T2DM Groups

ELISA was used to analyze serum adiponectin expression in the healthy population (control group, *n* = 8) and in patients with T2DM (T2DM group, *n* = 32). There were differences in adiponectin protein expression between the control and T2DM group (*p* = 0.037) (Figure 4A).

### 2.5. Serum Adiponectin Protein Expression in the Control and T2DM Groups (<5 Years and >15 y Disease Course)

Based on the ELISA analysis of the adiponectin protein expression in the healthy population (control group, *n* = 8) and in patients with T2DM (T2DM group, *n* = 32), the T2DM group was sub-divided into two subgroups, one with a disease course of <5 y (<5 y T2DM group, *n* = 16) and another, >15 y (>15 y T2DM group, *n* = 16). Compared with the control group, the adiponectin protein expression was lower in both T2DM sub-groups (disease courses < 5 y and >15 y) (<5 y T2DM group, *p* = 0.049; >15 years T2DM group, *p* = 0.032). Furthermore, the level of adiponectin protein expression appeared to decrease with increasing disease course in the T2DM group (Figure 4B).

## 3. Discussion

In this study, BKS.Cg-Dock^7m^ +/+ Lepr^db^/Jnarl (db/db) obese T2DM mice were used as animal models, as the T2DM pathological process is similar to humans. The adiponectin protein expression in the liver tissues of T2DM mice significant decreased compared to the control group (*p* < 0.05) in different disease courses (4, 16, and 32 w) (Figure 1). Furthermore, a similar result was also observed in the Figure 2. The IHC data demonstrated not only protein expression, but also subcellular localization of adiponectin protein in hepatic sinusoid of mice liver tissue (Figure 2).

The liver is an important organ for carbohydrate and lipid metabolism, and it is also the main organ that expresses adiponectin. Adiponectin binds to adiponectin receptors (AdipoR1 and AdipoR2), which in turn are activated to exert their biological effects [22]. In addition, it will stimulate the downstream adenylate-activated protein kinase (AMPK) signaling pathway, which can increase the uptake of glucose, promote fatty acid oxidation in cells, and inhibit the production of fatty acids in adipocytes [23,24]. However, low expression of AdipoR1 and AdipoR2 was observed in diabetic obese rats [18]. In the current study, the reduction of adiponectin in the liver seemed to be closely associated with T2DM-induced liver damage and lipid metabolism abnormalities.

Although more than 100 papers have been published on the adiponectin gene, only 30 are related to T2DM and only two are concerned with the SNPs selected in this study and the Taiwanese population. The relevant papers reported no significant differences in the SNPs of rs2241766 with respect to T2DM, but their sample sizes were relatively small [25]. By increasing the sample size in our study, rs2241766 was found to be correlated with T2DM, although rs2241766 T/G is a synonymous mutation that does not alter the amino acid sequence. A study has demonstrated the presence of a linkage disequilibrium between the rs2241766 polymorphism and rs266729. Since rs266729 is located in the 5′ promoter of the adiponectin gene, it can induce changes in adiponectin gene expression and plasma concentration [26]. Differences in rs2241766 allelic and genotypic frequencies among different ethnicities can lead to changes in the linkage disequilibrium structure, which may be the main reason for the variation in results across different ethnic groups. In addition, T2DM is a highly heterogeneous, polygenic disease that is also affected by environmental factors such as smoking, high-fat diets, and a lack of exercise. These factors may affect the correlation between adiponectin SNPs and adiponectin-associated risk of disease.

In addition to studying the adiponectin expression of liver tissues from T2DM mice, the level of serum adiponectin in patients with T2DM was also analyzed and appears to decrease with disease course. This is an interesting phenomenon that may be attributed to insulin regulating the secretion of various proteins from adipose tissue. Elevated plasma insulin in the diabetic subjects may have been responsible for the decreased plasma adiponectin concentrations. On the other hand, chronic insulin resistance in type 2 diabetes may be related to decreased plasma adiponectin. Decreased plasma adiponectin may play a causative role in the development of insulin resistance [27].

Adiponectin level can also be affected by certain factors such as obesity, age, blood lipids, gender, smoking status, blood glucose, renal function, adiponectin measurement method, genetic background, and drug treatment [28]. Adiponectin levels may be involved in the mechanism against inflammation, which may be a component of T2DM pathophysiology. Therefore, this study hypothesized that adiponectin is a protective factor against T2DM, whereby an increase in insulin resistance will cause the reduction of adiponectin, thereby regulating blood glucose changes in the body.

There are a few limitations to this study. First, the sample size for the measurement of human serum adiponectin level was relatively small, which meant that robust statistical data could not be obtained for the effects of adiponectin level on T2DM. Second, medication information was not collected from the patients, which may confound the relationship between serum adiponectin level and T2DM. Third, comorbidities are common in patients with T2DM, and may be treated using certain related drugs effect the expression level of adiponectin in serum [29].

To summarize, adiponectin may be correlated with insulin resistance [30]. Thus, we postulate that adiponectin is a protective factor against T2DM. As insulin resistance increases, it may become responsible for the decreased serum adiponectin concentrations and regulating changes in blood glucose in the diabetic patients.

## 4. Materials and Methods

### 4.1. Sample Collection

In this study, we recruited a total of 570 T2DM patients (63.6 ± 11.5 years and 51.4% male individuals) over 20 years of age at China Medical University Hospital (Taichung City, Taiwan) between August 2014 and July 2015. The diagnostic criteria for T2DM patients are described in detail in Ref. [31]. The genotype frequency data of 1700 healthy controls were downloaded from the Taiwan Biobank (https://taiwanview.twbiobank.org.tw/ accessed on 1 September 2016) (case no: TWBR10509-02; control no: TWBR10309-001) [32] to determine the prevalence of polymorphism in these patients. Informed consent was obtained from all analyzed patients.

The present study obtained the rs1063538 and rs2241766 single-nucleotide polymorphisms (SNPs) at chromosome region 3q27.3 in adiponectin (Figure 3) from the National Center for Biotechnology Information’s SNP database (http://www.ncbi.nlm.nih.gov/snp (accessed on 13 June 2017)). Tag SNPs were selected using the Tagger function (http://software.broadinstitute.org/mpg/tagger/server.html (accessed on 13 June 2017)) with the additional criteria: (i) A threshold minor allele frequency in the HapMap phase 3; and (ii) Han Chinese in Beijing, China (CHB) + Japanese in Tokyo, Japan (JPT) population of 0.05 for ‘tag SNPs’. All study protocols were approved by the Ethical Committee of China Medical University Hospital (approval no. CMUH103-REC2-071).

### 4.2. Animal Model

A total of 24 male (age, 4 weeks old) background BKS.Cg-Dock^7m^ +/+ Lepr^db^/JNarl (db/db) mice were purchased from the National Laboratory Animal Center. The control group mice (*n* = 12) with the +Dock^7m^/+Dock^7m^ genotype and those with the +Lepr^db^/+Lepr^db^ genotype constituted the T2DM group (*n* = 12). The present study involved animal experiments and considered the 3R principles of ‘Replace’, ‘Reduce’ and ‘Refine’ to optimize experimental design [33]. Procedural details are available in Ref. [11]. According to the information from Jackson Laboratory (JAX stock #000642) [34], the mouse model exhibited elevated blood sugar at 4–8 weeks and mortality by 10 months of age. In the present study, the mice were divided into six groups (four mice per group), namely three for control groups: early stage (4 weeks), middle stage (16 weeks), and late stage (32 weeks), and three for T2DM groups: early stage (4 weeks), middle stage (16 weeks), and late stage (32 weeks). Mice were sacrificed and the liver tissue was obtained at the scheduled time. Procedural details are available in Ref. [35]. The present study was reviewed and approved by the Institutional Animal Care and Use Committee (IACUC) of China Medical University (IACUC permit no. 2016-221).

### 4.3. Western Blot Analysis

The WB was used to detect adiponectin in the liver tissue of mice with murine anti- adiponectin (GTX23455) polyclonal antibody. The procedures of the WB have been described previously [36]. Briefly, primary anti-adiponectin antibodies (1:1000 dilutions) (GTX23455, GeneTex, CA, USA) were used to detect adiponectin. To control equal loading of total protein in all lanes, blots were stained with mouse anti-β-actin antibody at a 1:5000 dilution (cat. no. ab8226; Abcam, Cambridge, MA, USA). The bands were visualized using an ECL kit (GE Healthcare) according to the manufacturer’s protocol. Finally, the blots were visualized by enhanced chemiluminescence using an ImageQuant™ LAS 4000 system (GE Healthcare) [37].

### 4.4. Immunohistochemistry (IHC) Analysis

IHC was performed on paraffin-embedded liver sections. Adiponectin (ADP) immunohistochemistry staining was carried out by a Leica Bond MAX automated immunostainer (Leica Microsystems Inc., Buffalo Grove, IL, USA). IHC was performed as described previously [38,39]. The tissue sections were dewaxed, treated with Proteinase K enzyme, and followed by blocking with 3% hydrogen peroxide for 10 min. After washing in PBS (Phosphate Buffered Saline PH 7.6) 5 min, the slides were incubated in anti-adiponectin (GTX23455, GeneTex, CA, USA) antibody for 30 min, followed by rabbit anti-rat antibody, and then a goat anti-rabbit HRP polymer for 15 min. Chromogen visualization was performed using 3,3′-diaminobenzidine tetrahydrochloride (DAB) for 5 min. The sample was washed with 0.05% Tween 20/Tris-buffered saline (DAKO, Carpinteria, CA, USA) in order to perform all the necessary steps [40,41,42]. The adiponectin protein expression was defined using light microscopy (Leica DM 1000 LED Lab; cat. no. 10052-384; Leica Microsystems, Inc., Wetzlar, Germany) at a magnification of ×20 or ×200.

### 4.5. Enzyme-Linked Immuno-Sorbent Assay (ELISA)

In this study, thirty-two patients were enrolled at China Medical University Hospital (CMUH) in Taiwan and fulfilled the diagnostic criteria for obesity (BMI > 27) and T2DM. And eight gender-age-BMI matched unrelated healthy controls from the general population were selected through physical examination at CMUH in Taiwan.

Adiponectin concentrations in serum samples were analyzed using Human ADP (Adiponectin) ELISA Kit following the manufacturer’s protocol (Fine Test, EH2593) (Wuhan Fine Biotech Co., Ltd., Wuhan, China). Absorbance at 450 nm wavelength was read using an ELISA plate reader (Tecan Sunrise Reader, 96-well Microplate Reader). The concentrations of adiponectin in serum samples were determined by interpolation from the standard curve. All experiments were conducted in triplicate.

### 4.6. Statistical Analysis

Statistically significant differences in allele/genotype frequencies of adiponectin SNPs (rs1063538 and rs2241766) between the T2DM and control groups were determined using the χ2 test. Odds ratios were calculated from the genotypic frequency and allelic frequency at 95% CI for the adiponectin SNPs (rs1063538 and rs2241766). Statistical analysis was performed using SPSS software (version 11; SPSS, Inc., Chicago, IL, USA) Data from three independent experiments are expressed as the mean ± SE. Statistical comparisons between the T2DM and control groups were performed using Student’s *t*-test. *p* < 0.05 was considered to indicate a statistically significant difference.

## 5. Conclusions

A schematic diagram of this study is shown in Figure 5. In conclusion, adiponectin is a potential T2DM susceptibility gene, and it may participate in the pathogenesis of T2DM by regulating blood glucose and lipid metabolism. Further experiments are required to validate the results of this study and provide more comprehensive information to understand the potential mechanisms of adiponectin at different T2DM stages to offer a new direction for T2DM treatment in the future.

## Figures and Tables

**Figure 1 ijms-23-13544-f001:**
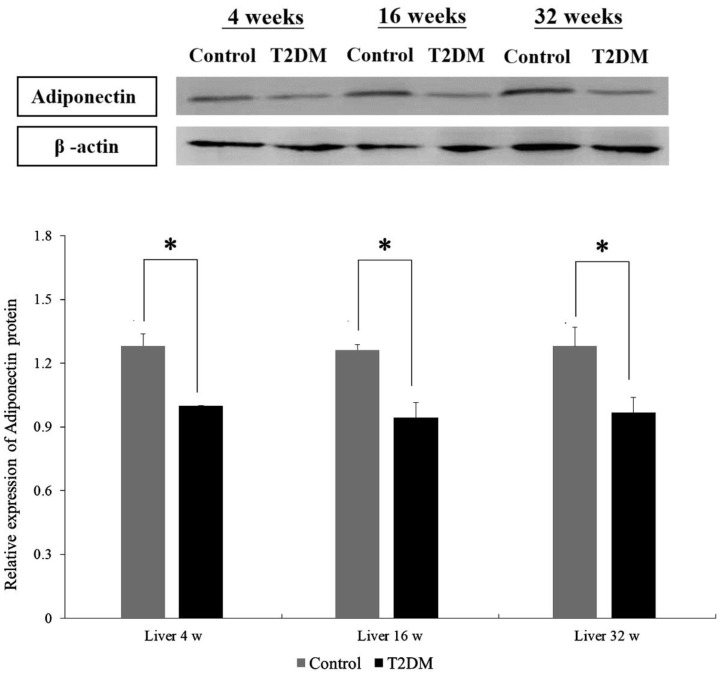
Western blot analyses of protein expression levels of Adiponectin (ADP) in the liver tissue of Control and T2DM mice at 4, 16, and 32 weeks, respectively. Data are expressed as mean ± standard error of 3 independent experiments. * *p* < 0.05 for the comparisons indicated.

**Figure 2 ijms-23-13544-f002:**
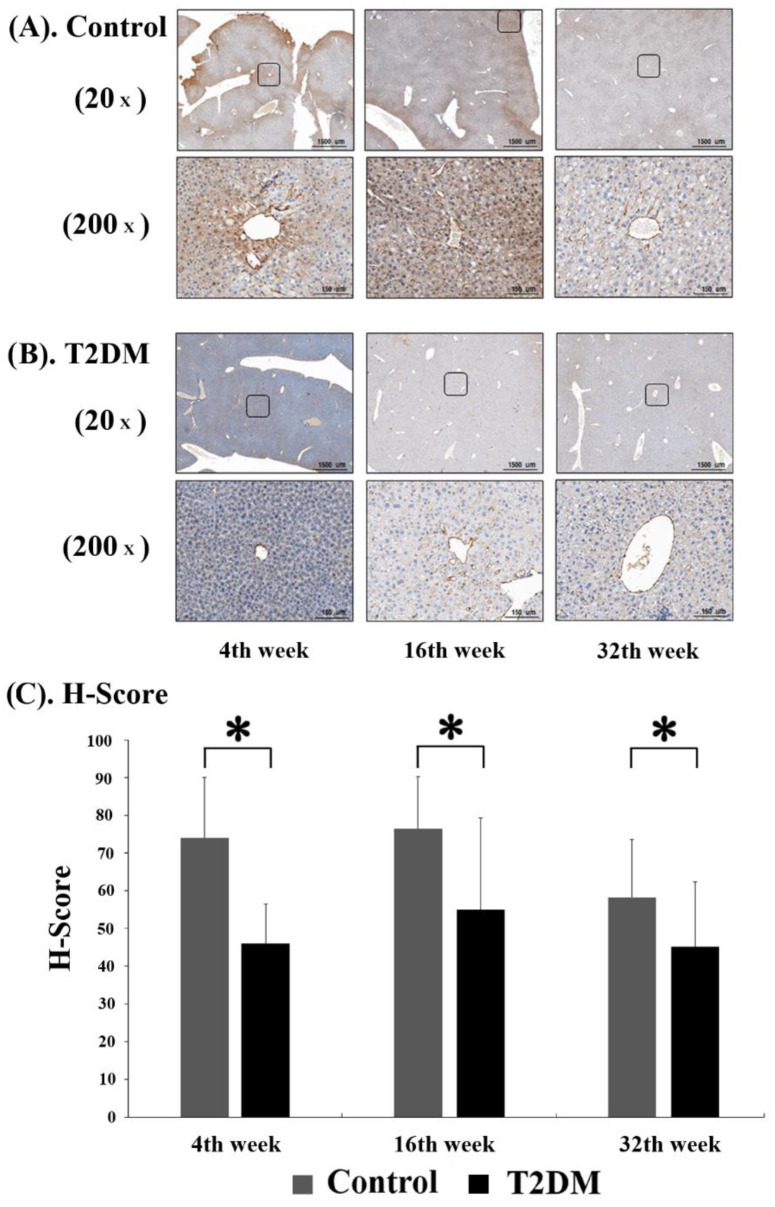
Immunohistochemistry (IHC) assay for Adiponectin (ADP) expression in control (**A**) and T2DM (**B**) mice was shown. The progression of mice model development for T2DM. Liver tissues were excised, fixed, paraffin-embedded, and sectioned for immunohistochemistry staining as described in Materials and Methods. Data are expressed as mean ± standard error of 3 independent experiments (**C**). * *p* < 0.05, for the comparisons indicated.

**Figure 3 ijms-23-13544-f003:**
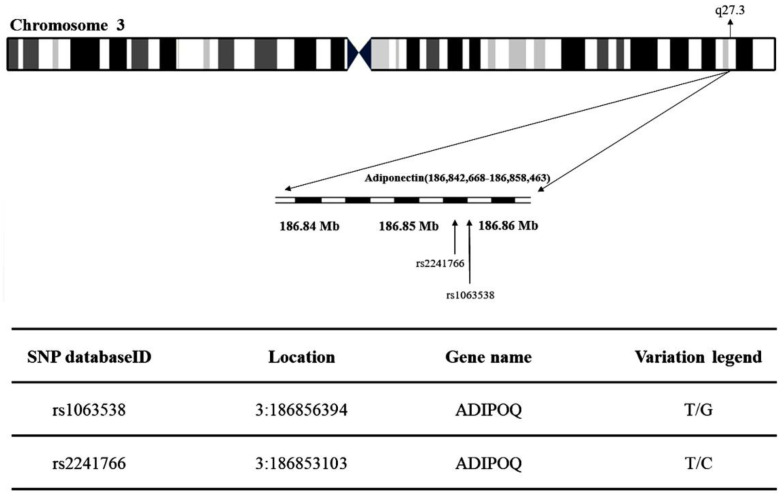
Map of adiponectin (rs1063538 and rs2241766) single-nucleotide polymorphisms (SNPs) located at chromosome region 3q27.3.

**Figure 4 ijms-23-13544-f004:**
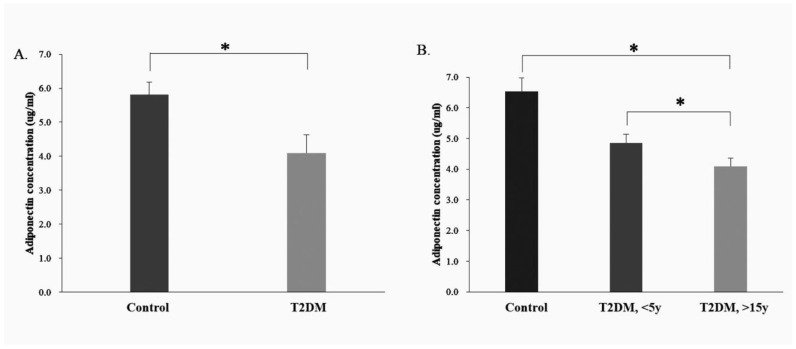
Measurement of the adiponectin protein concentrations in the serum of control group and T2DM patients using ELISA analysis. (**A**). Serum adiponectin expression in the human control and T2DM groups; (**B**). Serum adiponectin expression in the human control and different course of T2DM groups (<5 Years and >15 y Disease Course). * *p* < 0.05, for the comparisons indicated.

**Figure 5 ijms-23-13544-f005:**
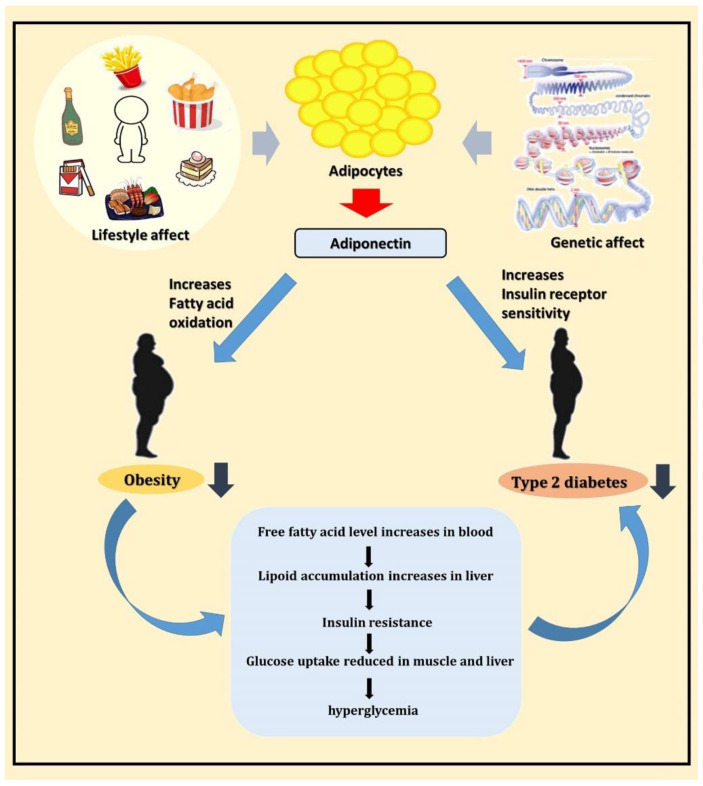
Schematic diagram showing the relationship between adiponectin, obesity, and T2DM.

**Table 1 ijms-23-13544-t001:** Genotypic and allelic frequencies of Adiponectin genetic polymorphisms in the patients with T2D and controls.

dbSNP ID		Patient with T2D	Control	OR (95% CI)	*p*-Value
		**(N = 570)**	**(N = 1700)**		
rs1063538					
	Genotype	**(N = 570)**	**(N = 1696)**		
	CC	102 (17.9)	285(16.8)	1.13 (0.86–1.50)	0.637
	CT	287 (50.4)	838 (49.4)	1.08 (0.88–1.34)	
	TT	181 (31.8)	573(33.8)	Ref	
	Allele frequency				
	C	491 (43.1)	1408 (41.5)	1.06 (0.93–1.22)	0.355
	T	649 (56.9)	1984 (58.5)	Ref	
rs2241766					
	Genotype	**(N = 566)**	**(N = 1695)**		
	GG	39 (6.9)	173 (10.2)	0.61 (0.42–0.89)	0.025 *
	GT	230 (40.6)	717 (42.3)	0.87 (0.71–1.06)	
	TT	297 (52.5)	805 (47.5)	Ref	
	Allele frequency				
	G	308 (27.2)	1063 (31.4)	0.82 (0.70–0.95)	0.009 *
	T	824 (72.8)	2327 (68.6)	Ref	

CI, confidence interval; OR, odds ratio; *, *p* < 0.05.

## Data Availability

The datasets used and/or analyzed during the current study available from the corresponding author on reasonable request.

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
