# Peer review of "Genetic and Functional Effects of Adiponectin in Type 2 Diabetes Mellitus Development"

_ijms, 2022, doi:10.3390/ijms232113544_

Round 1

Reviewer 1 Report

The article presents novel interesting data on the association of polymorphism of the ADIPOQ adiponectin gene with the incidence of diabetes 2. For the first time, the relationship between the mutation at the rs2241766 locus and T2DM was revealed. These data are of great importance in the diagnosis and prognosis of the course of T2DM and associated metabolic disorders.

Main notes: The feasibility of a combined study in a mouse model with diabetes and in humans is unclear. There is no convincing evidence for the need for a combination study. The introduction does not provide a clear explanation of the design of the study.

Human study:

- It has been known for over 20 years that a decrease in adiponectin is observed in diabetic patients (doi: 10.1161/01.atv.20.6.1595.). Therefore, this data is not new.

- Has the adiponectin level been adjusted for body weight?

Animal study:

- The origin of adiponectin, which has been detected in the liver of mice, is unclear - mRNA has not been determined.

- It is not clear why the expression of adiponectin in the liver of mice was studied both by the method of western blot and immunohistochemistry. A discussion of this issue should be added.

- Perhaps it would be better to study the expression of adiponectin in mice in adipose tissue.

Minor notes:

- Figure 1 «in the live tissue» should be change to «in the liver tissue»

- Figure 2 – absence of markers of significance; embed-ded should be change to embedded. There are extra dashes throughout the text of the article.

- Figure 4 – wrong subscriotion

- Discussion, 5 paragraph: «A study has demonstrated that insufficient insulin secretion in patients with T2DM can stimulate the adipocytes in adipose tissue, causing an increase in adipose secretion, which in turn enhances insulin sensitivity and regulates in vivo hyperglycemia».  A citation should be added.

- The bibliography is double numbered.

Resolution: may be published after revision. It may be better not to present animal and human data in one article.

Author Response

Reviewer # 1 Comments:

The article presents novel interesting data on the association of polymorphism of the ADIPOQ adiponectin gene with the incidence of diabetes 2. For the first time, the relationship between the mutation at the rs2241766 locus and T2DM was revealed. These data are of great importance in the diagnosis and prognosis of the course of T2DM and associated metabolic disorders.

Main notes:

The feasibility of a combined study in a mouse model with diabetes and in humans is unclear. There is no convincing evidence for the need for a combination study. The introduction does not provide a clear explanation of the design of the study.

Response:

We do appreciate the helpful comments. We had added some description in the section of the Introduction (page 4, line 71 to line 74 with the red color).

Human study:

- It has been known for over 20 years that a decrease in adiponectin is observed in diabetic patients (doi: 10.1161/01.atv.20.6.1595.). Therefore, this data is not new.

Response:

We would like to thank the reviewer for the comment. We know that a decrease in adiponectin is observed in diabetic patients in previous studies. However, we are the first one to discuss the expression level of adiponectin appeared to decrease with increasing disease course in the T2DM. We also the first one to define the relationship between rs2241766 SNP and T2DM.

- Has the adiponectin level been adjusted for body weight?

Response:

We do appreciate the reviewer for the comment. We had added some description in the section of Methods (page 12, line 267 to page 13, line 277 with the red color).

Animal study:

- The origin of adiponectin, which has been detected in the liver of mice, is unclear - mRNA has not been determined.

Response:

We would like to thank the reviewer for the comment. Actually, we did have gene expression data. As shown as below, the similar result was also observed in the expression level of mRNA. We had added some description in the section of Results (page 5, line 103 to 104 with the red color).

- It is not clear why the expression of adiponectin in the liver of mice was studied both by the method of western blot and immunohistochemistry. A discussion of this issue should be added.

Response:

We do appreciate the reviewer for the comment. We had added some description in the section of Discussion (page 7, line 154 to 159 with the red color).

- Perhaps it would be better to study the expression of adiponectin in mice in adipose tissue.

Response:

We do appreciate the reviewer for the comment. We will consider using adipose tissue for the Further investigations.

Minor notes:

- Figure 1 «in the live tissue» should be change to «in the liver tissue»

Response:

We do appreciate the reviewer for the comments. We had corrected the typo in the section of Figure Legends (page 22, Figure 1 legend with the red color).

- Figure 2 – absence of markers of significance; embed-ded should be change to embedded. There are extra dashes throughout the text of the article.

Response:

We would like to thank the reviewer for the comment. We had revised the Figure 2 and also change the typo in the section of Figure Legends (page 22, Figure 2 legend with the red color).

- Figure 4 – wrong subscriotion

Response:

We do appreciate the reviewer for the comments. We had revised the typo in the section of Figure Legends (page 22, Figure 3 and 4 legend with the red color).

- Discussion, 5 paragraph: «A study has demonstrated that insufficient insulin secretion in patients with T2DM can stimulate the adipocytes in adipose tissue, causing an increase in adipose secretion, which in turn enhances insulin sensitivity and regulates in vivo hyperglycemia».  A citation should be added.

Response:

We would like to thank the reviewer for the comment. We had revised the whole section of Discussion (page 7, line 154 to page 10, line 207 with the red color).

- The bibliography is double numbered.

Response:

We would like to thank the reviewer for the comment. We had rechecked the manuscript.

Reviewer 2 Report

The study by Tang et. al identifies the association of two SNPs in adiponectin gene with type 2 Diabetes. Below are some comments

1.     Specify number of samples (n) in Figure 1 and 2 used for quantification.

2.     Figure 1 legend “live tissue” instead of “liver tissue”.

3.     The text in result 2.3 about the genotypic frequencies of both SNPs in control and T2D groups are written opposite as compared to what is shown in the table.

4.     Figure legend written for figure 3 and figure 4 are for figure 4 and figure 3 respectively.

5.     The whole discussion section discusses the result in completely opposite sense, the authors found reduced levels of adiponectin in T2D mouse and humans, whereas the discussion explains as the adiponectin levels were higher in T2D and explains otherwise.

Author Response

Reviewer # 2 Comments:

The study by Tang et. al identifies the association of two SNPs in adiponectin gene with type 2 Diabetes. Below are some comments.

  1. Specify number of samples (n) in Figure 1 and 2 used for quantification.

Response:

We would like to thank the reviewer for the comments. We had some description in the section of Materials and Methods (page 11, line 228 to 229; line 235 to 238; page 13, line 284 to 286 with the blue color). We also added some description in the section of Figure Legends (page 22, Figure 1 and 2 legend with the red color).

  1. Figure 1 legend “live tissue” instead of “liver tissue”.

Response:

We do appreciate the reviewer for the comments. We had corrected the typo in the section of Figure Legends (page 22, Figure 1 legend with the red color).

  1. The text in result 2.3 about the genotypic frequencies of both SNPs in control and T2D groups are written opposite as compared to what is shown in the table.

Response:

We do appreciate the reviewer for the comments. We had revised the typo in the section of Results (page 6, line 115 to 135 with the red color).

  1. Figure legend written for figure 3 and figure 4 are for figure 4 and figure 3 respectively.

Response:

We do appreciate the reviewer for the comments. We had revised the typo in the section of Figure Legends (page 22, Figure 3 and 4 legend with the red color).

  1. The whole discussion section discusses the result in completely opposite sense, the authors found reduced levels of adiponectin in T2D mouse and humans, whereas the discussion explains as the adiponectin levels were higher in T2D and explains otherwise.

Response:

We would like to thank the reviewer for the comments. We had revised the whole section of Discussion (page 7, line 154 to page 10, line 208 with the red color).

Round 2

Reviewer 1 Report

All comments have been corrected, the necessary clarifications have been made.

Reviewer 2 Report

The authors modified the manuscript accordingly.